Transgenic LRRK2R1441G rats–a model for Parkinson disease?

Shaikh Komal T.
Yang Alvin
Youshin Ekaterina
Schmid Susanne susanne.schmid@schulich.uwo.ca
Department of Anatomy & Cell Biology, Schulich School of Medicine and Dentistry, University of Western Ontario , London, ON , Canada
Yuan Tifei
Electronic publication date: 2015 May 12
Publication date: 2015
Volume: 3
Electronic Location ID: e945
Received 2015 Jan 21; Accepted 2015 Apr 18
Copyright: © 2015 Shaikh et al.
Copyright year: 2015
Copyright holder: Shaikh et al.
License: This is an open access article distributed under the terms of the Creative Commons Attribution License, which permits unrestricted use, distribution, reproduction and adaptation in any medium and for any purpose provided that it is properly attributed. For attribution, the original author(s), title, publication source (PeerJ) and either DOI or URL of the article must be cited.
License URL: https://creativecommons.org/licenses/by/4.0/

Keywords: Parkinson disease, Transgenic rat, Animal model, Paraquat, Lrrk2, Motor testing, Cognitive function

Funding: Canadian Institute for Health Research This study was funded by the Canadian Institute for Health Research and by the Canadian Foundation for Innovation (CFI), Leaders of Excellent Funds. The funders had no role in study design, data collection and analysis, decision to publish, or preparation of the manuscript.

==============================
Parkinson disease (PD) is the most common movement disorder, characterized by the progressive degeneration of dopaminergic neurons in the substantia nigra. While the cause of this disease is largely unknown, a rare autosomal dominant familial form of PD is caused by a genetic mutation in the leucine-rich repeat kinase 2 (LRRK2) gene that presumably leads to a gain-of-function of LRRK2 kinase activity. Here, we explored the potential of over expression of this human gene in a new transgenic rat model to serve as an animal model for PD. Commercially available BAC transgenic rats expressing human LRRK2 with the familial PD mutation, R1441G, and their wild-type siblings were tested for deficits in motor function, sensorimotor gating, and higher cognitive function reminiscent of PD through the ages of 3, 6, 9 and 12 months. At 12 months of age, rats were exposed to intraperitoneal injections of the environmental toxin Paraquat or saline. Our results indicate that LRRK2R1441G transgenic rats do not show signs of neurodegeneration and do not develop significant motor or cognitive deficits until the age of 16 months. In addition, LRRK2R1441G transgenic rats did not show increased vulnerability to sub-toxic doses of Paraquat. Gene expression studies indicate that despite genomic presence and initial expression of the transgene, its expression was greatly reduced in our aged rats. We conclude that the transgenic LRRK2R1441G rat is not a valid model for studying the pathology of PD and discuss this in relation to other transgenic rat models.

Introduction

Parkinson disease (PD) is characterized by the degeneration of dopaminergic neurons in the substantia nigra pars compacta (SNpc) causing cardinal motor symptoms including resting tremor, rigidity, bradykinesia, and abnormal gait. These symptoms occur relatively late in the time course of the disease by the time when 60% of dopaminergic neurons have degenerated and striatal dopamine (DA) content has been reduced by 80% (Bernheimer et al., 1973). In addition, PD patients may present with cognitive symptoms, including depression, anxiety and impaired memory (Emre, 2003).

While most cases of PD are sporadic, 5–10% are associated with familial mutations, most commonly in the LRRK2 gene (Paisan-Ruiz et al., 2004; Zimprich et al., 2004). LRRK2 encodes leucine-rich-repeat kinase II, a large multidomain protein with both kinase and GTPase enzymatic functions (Zimprich et al., 2004; Santpere & Ferrer, 2009). In addition, LRRK2 has several functional motifs and has been implicated in a variety of cellular processes including mitochondrial function, signal transduction, cell death pathways, vesicle trafficking, neurite outgrowth, autophagy and cytoskeleton assembly (Santpere & Ferrer, 2009; Cookson, 2010; Berwick & Harvey, 2011; Tsika & Moore, 2012). The R1441G mutation on LRRK2 decreases GTPase activity and might thereby increase kinase activity, however, the physiological function of LRRK2 and the regulation of its enzymatic activity is not fully understood (Healy et al., 2008; Tsika & Moore, 2013). It is the second most common mutation causing PD, with a progression that is indistinguishable from sporadic PD, suggesting similar underlying mechanisms.

Transgenic mouse models were developed to gain insight into the mechanisms through which familial LRRK2 mutations are linked to PD pathogenesis. Li and colleagues (2009) described a LRRK2R1441G BAC transgenic mouse line that recapitulates human PD phenotypes, including L-DOPA responsiveness and age dependent motor deficits starting at 6 months of age and progressively worsening by 12 months of age. While these animals did not show dopaminergic neuron degeneration in the SNpc or aggregation of α-synuclein, dopamine release was impaired and axonal dystrophy reminiscent of PD was noted in the striatum. However, following studies with these mice have largely failed to reproduce the motor dysfunction, only mild Parkinsonism, gastrointestinal dysfunction, and impaired dopaminergic transmission were reported in some studies (Bichler et al., 2013; Dranka et al., 2013).

One explanation for the lack of full PD pathology in the transgenic mouse may be that multiple factors, such as interactions with other genes or environmental stressors are required to inhibit compensatory mechanisms and facilitate the degenerative process. The relatively low incidence of familial PD, failure to recapitulate PD phenotypes in genetic models, and lack of singular environmental insults causing PD, has prompted the emergence of the ‘multiple hit’ hypothesis of PD. It suggests that multiple risk factors interact to induce the degenerative process, with the primary insult causing cellular stress and all succeeding insults resulting in a loss of protective pathways which together lead to neuronal death (Sulzer, 2007). Exposure to environmental toxins, particularly agrochemicals, has been suggested as possible additional risk factor. In particular, exposure to N, N′-dimethyl-4-4′-bipyridinium dichloride, or Paraquat, has been shown to increase PD risk (Costello et al., 2009). Paraquat, a widely used herbicide, induces PD related neuropathology presumably through increased oxidative stress and production of reactive oxygen species (ROS).

The aim of the present study was to determine if mutated LRRK2 induces PD phenotypes in a new transgenic rat model, BAC LRRK2R1441G rats, and how exposure to Paraquat influences the phenotype.

Materials and Methods

Animals

A commercially available breeding pair of Sprague Dawley rats expressing the R1441G mutation on the human gene LRRK2 was obtained from Taconic (Germantown, New York, USA). According to the Taconic records and qRT-PCR results published on the Taconic website at the time of purchase, hemizygous rats expressed 5–10 copies of the transgene. The model was originally created by Dr. Chenjian Li, with support of the Michael J. Fox Foundation, through pronuclear injection of the human LRRK2R1441G gene into Sprague Dawley zygotes. Animals for this study were obtained through in house breeding of the purchased pair which was a hemizygous male and a wild-type SD female. Pubs were weaned at 3 weeks of age and genotyped to detect human LRRK2. Transgenic animals were housed with wild-type littermates in a 12 h light-dark cycle with food and water provided ad libitum. All 17 wild-type and 24 LRRK2R1441G transgenic rats underwent the entire battery of behavioral tests unless otherwise noted. Animals underwent behavioral tests at 3 months, 6 months, 9 months and 12 months of age. All procedures were in accordance with the ethical guidelines of the Canadian Council on Animal Care (CCAC) and approved by the University of Western Ontario Animal Use Subcommittee, protocol no. 2011-077.

Genotyping

Before weaning, all rats were genotyped by polymerase chain reaction (PCR) using tissue obtained from ear-punching. Genotyping was performed using an assay kit from Taconic according to their specifications. The PCR reaction combined 5 µL of genomic DNA (2 ng/µL), 2.5 µL of PCR Buffer (5 mM), 1 µL of MgCl2 (2.5 mM), 0.5 µL of deoxynucleotide mixture (0.2 mM), 0.5 µL of hpark8-F primer (0.5 µM), 0.5 µL of hpark8-R primer (0.5 µM) and 0.25 µL of HotStarTaq DNA Polymerase (0.05 U/µL). The thermocycler protocol involved 15 min at 95 °C, 35 cycles with 45 s at 94 °C, 1 min at 65 °C, and 1 min at 72 °C, and 5 min. extension at 72 °C. The primer sequence for the hpark8-F primer was GAT AGG CGG CTT TCA TTT TTC C and for the hpark8-R primer ACT CAG GCC CCA AAA ACG AG. Primers were generated in house at the UWO Oligo factory (see Fig. S1). All rats were genotyped for a second time post-mortem.

Behavioral testing

All 17 wild-type and 24 LRRK2R1441G transgenic rats underwent the entire battery of behavioral tests at the different age stages unless otherwise noted. Tests were always performed in the same order and on separate days.

Vibrissae-evoked forelimb placing

The forelimb placing test was performed as previously described and used as a measure of movement initiation abilities (Woodlee et al., 2005), a common deficit observed in PD patients. Briefly, the experimenter holds the animal aloft by its torso and brushes its vibrissae against the edge of the testing surface. This elicits a forelimb placing response from the limb on the same side. Placing is quantified as the percentage of correct responses or ‘hits’ elicited out of fifteen trials. Trials in which the animal struggles or resists the experimenter’s grip are discounted. Animals were all trained on this task prior to testing in order to ensure acclimation to the experimenter as well as reduction in struggling behaviors.

Adjusting steps

The adjusting steps task has been extensively used to measure postural stability in rats (Olsson et al., 1995; Chang et al., 1999; Fleming, 2009). The experimenter holds the animal by its torso such that its hindlimbs are lifted above the testing surface. One forelimb is then restrained so that the animal’s weight is entirely supported by the remaining free forelimb, which is in contact with the testing surface. The experimenter then moves the animal laterally across a testing surface with a distance of 70 cm. In order to compensate for the movement of the body, the animal should make adjusting steps with the weight-bearing forelimb. The average number of steps each animal makes over five trials is recorded and used for analysis. Animals with nigrostriatal degeneration will drag their forelimb instead of making the appropriate adjusting steps (Fleming, Schallert & Ciucci, 2012).

Footprint analysis

Footprint analysis was performed in order to assess stepping patterns and abnormal gait (Li et al., 2010). The rat’s paws were dipped in non-toxic paint and it was placed on a runway 110 cm long, 10 cm wide and with 25 cm high side walls. The runway led to the rat’s home cage and rats were thus motivated to traverse the gangway. The floor of the runway was lined with ordinary paper and the footprints marked on the paper were analyzed. The distance between forepaws and the left and right stride were recorded.

Open field test

The open field test was used to measure general locomotor activity in wild-type and transgenic rats. Each animal was placed in a square activity box (Med Associates Activity Monitor, St. Albans, Vermont, USA) for 30 min per day, for two days. Horizontal and vertical movements were detected by two rows of photo beams. After each animal finished testing, the activity boxes were cleaned with 70% ethanol to eliminate any odors which may bias the next animal. Using the MedAssociates Activity Monitor software, we analyzed the distance traveled and the number of rearing movements during the testing interval. Exploratory rearing was used as an indirect measure of paucity of movement, as previously shown (Landers, Kinney & van Breukelen, 2014)

Acoustic startle response and sensorimotor gating test

The acoustic startle response of these animals was tested using a protocol similar to those previously described (Valsamis & Schmid, 2011; Typlt et al., 2013). Startle testing was conducted in sound attenuated startle boxes from Med Associates (MED ASR PRO1, St Albans, Vermont, USA). Animals were placed in holders mounted on a movement sensitive platform within the startle box. A transducer converted the vertical movement of the platform, induced by the animal’s startle response, into a voltage signal. The maximum amplitude of the signal was measured using Med Associates software (Startle Reflex Version 6, Med Associates, St Albans, Vermont, USA). On day 1, animals were acclimated to the startle box and background noise (65 dB white noise) for 5 min in the morning and again for 5 min several hours later. On day 2, animals underwent an input–output (IO) test to determine the appropriate gain setting for each animal. The IO function has an initial stimulation at 65 dB (20 ms duration) and increased in 5 dB intervals to 120 dB. On day 3, animals were tested in two blocks for short term habituation and prepulse inhibition respectively. Block 1 assessed habituation by presenting 30 trials of the startle pulse (105 dB white noise, 20 ms duration and 15 ms intertribal interval). Block 2 assessed prepulse inhibition. In Block 2, there were seven different trial conditions (10 trials per condition) for a total of 70 trials. The trials presented the startle pulse alone, a low prepulse of 75 dB (4 ms duration) before the startle pulse, and a high prepulse of 85 dB (4 ms duration) before the startle pulse. The presentation of the prepulse and the startling pulse was separated by three different interstimulus intervals (ISIs): 10 ms, 30 ms or 100 ms. The trials were presented in pseudorandomized order. Animals were tested on both habituation and prepulse inhibition as both responses are disrupted in PD patients (Matsumoto et al., 1992; Zoetmulder et al., 2014).

Morris water maze

Both wild-type and transgenic animals underwent two versions of the Morris water maze task to assess learning and memory. The first was a cued version that heavily relies on striatal function, the second was the classical spatial version that tests hippocampal function. The task was conducted in a round tank, 146 cm in diameter and 58 cm deep (Stoelting Co., Wood Dale, Illinois, USA), filled with water. The water was colored with non-toxic blue paint to ensure opaqueness. Throughout testing, the water temperature was monitored and maintained at 21 °C. The tank was divided into four equally sized quadrants and a circular acrylic escape platform was placed in one of the quadrants. The escape platform was submerged in water by 2 cm so that it was not visible to the animals. A camera mounted above the tank recorded the movement of the animals in each trial. The Any-maze Behavior Tracking Software (Stoelting Co., Wood Dale, Illinois, USA) was used to record the latency to reach the escape platform and the time spent in the target quadrant.

All animals were first tested on the cued version of the water maze task. This consisted of two training days with four trials on each day. In each trial, the animals were placed in the water facing the tank wall and had to locate the escape platform, which was cued by a yellow ball attached to the platform and protruding from the water. The trial was completed when the animal either found the escape platform or 90 s had passed. If the animal was unable to locate the platform in 90 s, it was gently led to the platform. Animals were allowed to remain on the escape platform for 15 s before being removed and dried before the next trial. The initial position of the animal was the vertices of one of the four quadrants and was different for each trial. The initial position was assigned randomly and counterbalanced for the genotypes. The platform position was also changed between each trial and was randomly assigned and counterbalanced.

The animals were also tested on the spatial reference version of the water maze as previously described (Miyoshi et al., 2002). This consisted of four training days with four trials on each day. The experimental procedure was similar to the previous one, except that the location of the platform was no longer cued. Instead, animals could utilize external visual cues on the walls surrounding the tank to locate the platform. In addition, the platform position was kept constant between trials and days. The first four trials of the first of the four training days were used as an indicator of spatial working memory.

On day 7, experimenters ran a 90 s probe trial without the platform. During this trial, the time spent in the target quadrant was recorded for each animal.

Paraquat injections

All animals that reached the age of 12 months were exposed to an acute sub-toxic Paraquat regimen at the age of 12 months, as previously described by Hutson et al. (2011), in order to assess their vulnerability to the environmental toxin. Animals were separated into four groups: wild-type/saline, wild-type/Paraquat, transgenic/saline, and transgenic/Paraquat. Within the genotypes, animals were randomly assigned to saline versus paraquat groups. Animals received two IP injections of Paraquat (10 mg/kg dissolved in 0.9% saline) or saline, with three days in between injections. At the time of last behavioural testing, animals were 14–18 months old. The toxin dose used in this study was a quarter of that previously shown to cause dopaminergic cell death in the substantia nigra in adult rats (Cicchetti et al., 2005). All animals underwent the open field test (for detailed description see above) immediately after receiving injections and 24 h after injections. As previously mentioned, MedAssociates Activity Monitor software was used to analyze the number of rearing movements during a 30 min testing interval.

Histology

One wild-type and one LRRK2R1441G rat of 3 months, 6 months, and 9 months of age and two rats per genotype of 18 months of age were used for histological analysis. Rats were deeply anaesthetized with sodium pentobarbital (54.7 mg/kg body weight; Comparative Medicine Animal Resources Center, Montreal, Ontario, Canada) and perfused transcardially with 0.9% saline solution at room temperature, followed by 0.1 M phosphate buffered saline (PBS; pH7.4) containing 4% paraformaldehyde (PFA) at 4 °C. Brains were then fixed in 4% PFA for 1 h and placed in 30% sucrose phosphate buffer (PB) at 4 °C until needed. Brain sections (40 µm thickness) containing the substantia nigra were cut. Free-floating brain sections were blocked in 1% H2O2 (30% stock solution) in 0.1M PBS (pH7.35), followed by treatment with PBS+(0.4% Triton-X and 0.1% BSA) for one hour. Sections were incubated with mouse anti-TH (1:8000) or rabbit anti-DAT (1:250, both Millipore, California, USA) in PBS+overnight. Next, sections were rinsed using PBS and incubated with biotinylated goat-anti-mouse or goat-anti-rabbit secondary antibodies in PBS+(1:500; from Vector Laboratories Inc., Burlingame, California, USA) for one hour. Thereafter, sections were rinsed with PBS, incubated with Avidin/Biotin Complex (1:1000; Vector Laboratories Inc., Burlingame, California, USA) for one hour; rinsed and then stained with 3-3′-diaminobenzidine (Sigma-Aldrich, Seelze, Germany) in 0.1M PB with 0.0004% H2O2. All incubations were done at room temperature. Sections were then rinsed with PB and mounted using 0.3% Gelatin A, and air-dried. Some TH sections were processed for Nissl counterstaining using a standard protocol, and cover slipped using Entellan mounting medium (EM Science, Gibbstown, New Jersey, USA). For Hoechst 33342 staining, free-floating brain sections were treated with Thermo Scientific Hoechst 33342 fluorescent stain according to (Thermo Fisher Scientific, Waltham, Massachusetts, USA), rinsed with PBS and mounted using 0.3% Gelatin A and air-dried. Slides were cover slipped using Vectrashield (Vector Laboratories Inc., Burlingame, California, USA). TH stained sections were examined using a Zeiss Axioplan microscope and pictures were taken at 2.5X magnification for quantification. Hoechst stained cells were counted under 63Xs magnification. For each brain area of interest, 3 sections per animal were matched between wild- type and experimental cohorts for examination and calculation of averages. Cells stained for TH were quantified using ImageJ. Section images were adjusted to 8-bit black and white and threshold was set to white = 0 and black = 225. The mean grey value was obtained to indicate the level of staining per section. Section labels were coded and covered during analysis so that the experimenter was blinded to the experimental group, in order to prevent group bias. All sections were analyzed by two persons independently.

q-RT PCR

Surviving transgenic rats were perfused after the Paraquat study with saline and samples for various brain regions (cortex, substantia nigra, hippocampus) as well as liver and kidney were obtained and frozen on ice. RNA was isolated using QIAzol Lysis Reagent (Qiagen, Germantown, Maryland, USA) according to the manufacturer’s instructions. cDNA was synthesized using Life Technologies high-capacity cDNA Reverse Transcription Kit (Life Technologies, Grand Island, New York, USA). Real time PCR assays were performed in triplicate on a 384 well plate. The level of human LRRK2 mRNA was detected using TaqMan probe Hs00411197_ml specific for human LRRK2 (Life Technologies, Grand Island, New York, USA). The housekeeping gene GAPDH was detected using Rn01775763 (Life Technologies, Grand Island, New York, USA) and was used as a reference gene.

Statistical analysis

Mean values ± standard error are reported. Outliers, defined as data points three standard deviations from the mean, were identified and removed from the data set. Comparisons of genotype and treatment groups were performed using repeated measures ANOVA. The Huynh–Feldt correction was used when sphericity was violated in behavioural experiments and epsilon was greater than 0.75. When sphericity was violated but epsilon was lower than 0.75, the more conservative Greenhouse-Geisser correction was used to adjust the degrees of freedom. IBM SPSS Statistics 2.0 software was used for all statistical analyses. Results were considered statistically significant at a p value of 0.05.

Results

All animals were weighed before behavioral testing, at each age point. All rats significantly increased their weight over time (ANOVA F(2, 75) = 378,90, p < 0.001, Greenhouse-Geisser correction), however there was no significant difference in weight between transgenic LRRK2R1441G rats and their wild-type littermates (ANOVA F(2, 75) = 0.63, p = 0.52, Greenhouse-Geisser correction; data not shown).

Motor tests

In order to assess movement initiation abilities, vibrissae-evoked forelimb placing responses were measured in transgenic LRRK2R1441G rats and their wild-type littermates at 3, 6, 9 and 12 months of age. While a significant main effect of age was noted in forelimb placing responses (ANOVA F(2, 85) = 8.14, p < 0.05, Huynh–Feldt correction), no genotype and age interaction was noted (ANOVA F(2, 85) = 0.37, p = 0.71, Huynh–Feldt correction). Therefore, transgenic LRRK2R1441G did not significantly differ from wild-type rats in vibrissae-evoked forelimb placing, suggesting intact movement initiation abilities (Fig. 1A).

Figure 1 Motor testing.

(A) Vibrissae evoked forelimb placing was measured in transgenic LRRK2R1441G rats and wild-type littermates. While there was a significant influence of age group, there was no significant difference in forelimb placing between wild-type and transgenic LRRK2R1441G rats at 3, 6, 9, and 12 months of age. (B) Performance in the adjusting steps task revealed also no significant differences in the number of adjusting steps between transgenic LRRK2R1441G rats and wild-type littermates at 3, 6, 9, and 12 months of age, while there was a significant of age group. (C) Stride length was measured for each side (L indicates left side and R indicates right side) in each animal and averaged within genotype. Stride length increased significantly with age, but there was no significant difference in stride length in transgenic LRRK2R1441G rats and wildtype littermates at 3, 6, 9, and 12 month of age. (D) Total distance travelled during a 30 min. session in the locomotor box. While there was again a significant influence of age group, there was no significant difference in total distance travelled between transgenic LRRK2R1441G rats and wild-type littermates. (E) Rearing movements made by all animals was counted during testing sessions in the locomotor boxes. There was a significant decrease of rearing as animals aged, however, no significant difference was noted between transgenic LRRK2R1441G rats and wild-type littermates. (F) Maximal velocity of transgenic and wild-type rats was calculated across testing sessions. No significant difference in maximal velocity was noted between transgenic LRRK2R1441G rats and wildtype littermates. N = 24 transgenic and n = 17 wild-type animals in all age groups.

Postural stability in was measured using the adjusting steps task. While a small but significant main effect of age was noted in the number of adjusting steps made (ANOVA F(3, 102) = 3.40, p < 0.05, Huynh–Feldt correction), no genotype and age interaction was noted (ANOVA F(3, 102) = 1.06, p = 0.37, Huynh–Feldt correction). Transgenic LRRK2R1441G rats did not significantly differ from wild-type rats in the adjusting steps task, suggesting normal postural stability (Fig. 1B).

Foot print analysis was conducted on all animals in order to assess gait patterns. A main effect of age was noted in stride length of animals (ANOVA F(3, 105) = 37.36, p < 0.05, Huynh–Feldt correction). Both right and left stride lengths were measured in all animals, however, no main effect of side was noted (ANOVA F(1, 39) = 0.12, p = 0.72, Huynh–Feldt correction). In addition, there was no interaction between age, side and genotype (ANOVA F(3, 117) = 0.60, p = 0.62, Huynh–Feldt correction).Therefore LRRK2R1441G rats showed normal gait patterns compared to wild-type controls (Fig. 1C).

Lastly, general locomotor activity of transgenic LRRK2R1441G rats and wild-type controls was assessed in the open field test. Animals were tested for two 30 min sessions across two consecutive days. Data presented here is calculated across both sessions. There was no significant difference between transgenic and wild-type animals in the total distance travelled during the two sessions (ANOVA F(3, 102) = 0.47, p = 0.67, Huynh–Feldt correction,), although a main effect of age group was noted (ANOVA F(3, 102) = 29.71, p < 0.001, Huynh–Feldt correction; Fig. 1D). Overall, it seems as rats became familiar with the testing boxes due to repeated exposure, exploratory locomotor behavior decreased. Similarly, no significant difference in rearing behavior was noted between transgenic and wild-type rats (ANOVA F(3, 103) = 0.79, p = 0.49, Huynh–Feldt correction), although an effect of age stage was noted (ANOVA F(3, 102) = 19.33, p < 0.001, Huynh–Feldt correction; Fig. 1E). The number of rearings dropped significantly in animals of 12 months of age. Finally, there was no difference in maximal velocity of transgenic LRRK2R1441G rats and wild-type controls in the locomotor boxes (ANOVA F(3, 117) = 0.59, p = 0.62; Fig. 1F).

In summary, while many motor tests showed general effects of age and/or repeated testing, there were no signs of any motor impairment in LRRK2R1441G rats up to the age of 1 year.

Sensorimotor gating tests

The acoustic startle response was used to investigate sensorimotor gating mechanisms, including habituation and prepulse inhibition. Baseline startle amplitude was calculated from the average of the first two startle responses of testing at each age stage and was not significantly different between transgenic and wild-type rats or between age stages (ANOVA F(2, 103) = 0.32, p = 0.79, Huynh–Feldt correction; data not shown). Habituation scores were calculated by dividing the average of the startle responses of trials number 25–30 by the maximum of the first three startle responses. Habituation scores of transgenic LRRK2R1441G rats and wild-type littermates did not significantly differ (ANOVA F(3, 102) = 0.33, p = 0.78, Huynh–Feldt correction; Fig. 2A). The time course of habituation was also assessed by examining the attenuation of startle response over the entire 30 trials using repeated measures ANOVA. At 3 months of age, a significant decrease in startle amplitude was noted (ANOVA F(16, 643) = 3.29, p < 0.001, Huynh–Feldt correction), however there was no significant difference between transgenic and wild-type rats (ANOVA F(16, 643) = 0.68, p = 0.82, Huynh–Feldt correction; Fig. 2B). Similarly, significant habituation was observed in all animals at 6 (ANOVA F(12, 476) = 2.85, p < 0.001, Huynh–Feldt correction), 9 (ANOVA F(16, 640) = 1.91, p < 0.05, Huynh–Feldt correction) and 12 months of age (ANOVA F(16, 637) = 2.97, p < 0.001, Huynh–Feldt correction; Fig. 2B), but no significant difference between the genotypes was observed (6 months ANOVA F(12, 476) = 0.78, p = 0.67, Huynh–Feldt correction; 9 months ANOVA F(16, 640) = 1.04, p = 0.42, Huynh–Feldt correction; 12 months ANOVA F(16, 637) = 0.92, p = 0.55, Huynh–Feldt correction).

Prepulse inhibition was measured using two different prepulse levels (75 dB and 85 dB) and three different interstimulus intervals (10 ms, 30 ms, 100 ms). At 3 months, a significant main effect of prepulse (ANOVA F(1, 39) = 32.10, p < 0.001) and ISI was noted (ANOVA F(2, 78) = 10.28, p < 0.001), however there was no significant difference between transgenic and wild-type rats (ANOVA F(2, 78) = 0.64, p = 0.53, Fig. 2C).

Therefore at 3 months of age, all rats displayed prepulse inhibition, with no significant difference between the two genotypes. Similarly, prepulse inhibition was noted at 6 months (ANOVA F(1, 39) = 61.53, p < 0.001, Huynh–Feldt correction; data not shown), 9 months (ANOVA F(1, 39) = 30.40, p < 0.001, Huynh–Feldt correction; data not shown), and 12 months of age (ANOVA F(1, 39) = 29.83, p < 0.001, Huynh–Feldt correction; Fig. 2C), however there was no difference in the extent of inhibition between transgenic LRRK2R1441G rats and wild-type controls (6 months ANOVA F(2, 61) = 0.228, p = 0.74, Huynh–Feldt correction; 9 months ANOVA F(2, 78) = 1.49, p = 0.23, Huynh–Feldt correction; 12 months ANOVA F(2, 78) = 1.59, p = 0.21, Huynh–Feldt correction).

Figure 2 Sensorimotor gating.

(A) Habituation scores were calculated for each rat by dividing the average of the last five startle responses by the maximum of the first three startle responses and then averaged across genotype. Habituation scores did not differ between transgenic LRRK2R1441G rats and wild-type littermates at 3, 6, 9, or 12 months of age. (B) Mean habituation curves for the age of 3 and 12 months. Both wild-type and transgenic rats showed a significant decrease of startle response amplitudes over 30 trials with no significant difference between genotypes. (C) Prepulse inhibition of startle in with a 75 dB and a 85 dB prepulse in 3 months and 12 months old rats. The remaining startle amplitude in relation to the baseline startle without any prepulse is plotted. There are no significant differences between genotypes or between age groups. N = 24 transgenic and n = 17 wild-type animals in all age groups.

In summary, LRRK2R1441G rats showed no signs of sensorimotor gating deficits at any age stage.

Morris water maze

At older age stages of 9 and 12 moths, all rats were additionally tested on the Morris water maze in order to assess spatial learning. In the cued version of the test, latency to find the escape platform improved over training days at 9 months (ANOVA F(1, 39) = 31.73, p < 0.001, Huynh–Feldt correction, Fig. 3A) and 12 months of age (ANOVA F(1, 38) = 7.11, p < 0.05, Huynh–Feldt correction, Fig. 3A). However, at both 9 months (ANOVA F(1, 39) = 0.06, p = 0.81, Huynh–Feldt correction) and 12 months (ANOVA F(1, 38) = 1.54, p = 0.22, Huynh–Feldt correction), there was no significant difference between transgenic LRRK2R1441G rats and wild-type controls (Fig. 3A).

Figure 3 Morris water maze.

(A) Latency to find the target platform was measured across two training days in the cued version of the Morris water maze task. Performance on this task improved from training day 1 to training day 2 in 9 month old rats (left) and 12 months old rats (right), however, there was no significant difference between transgenic LRRK2R1441G rats. (B) Spatial learning in the Morris Water maze task. The average latency until finding the platform of four trials per day is plotted for four consecutive training days at 9 months of age (left) and 12 months of age (right). Both genotypes show similar learning curves. (C) The first four trials of the first spatial reference training day in the Morris Water maze were analyzed separately to test for differences in working memory at 9 months (left) and 12 months (right) of age. There was no significant difference between genotypes. (D) On a final probe day, the platform was removed and the amount of time spent in the target quadrant was assessed for 9 months (left) and 12 months (right) old animals. Again, there was no significant difference between the genotypes. N = 24 transgenic and n = 17 wild-type animals in all age groups.

In the spatial version of the test, training improved the mean latency to the platform in all rats at 9 months (ANOVA F(2, 66) = 13.6, p < 0.001, Huynh–Feldt correction, Fig. 3B) and 12 months of age (ANOVA F(3, 102) = 15.34, p < 0.001, Huynh–Feldt correction, Fig. 3B), however no significant genotype difference was noted at either age point (9 months: ANOVA F(2, 66) = 0.93, p = 0.39, Huynh–Feldt correction; 12 months: ANOVA F(3, 102) = 1.09, p = 0.35, Huynh–Feldt correction; Fig. 3B).

In the working memory analysis of the test (first 4 trials of the first training day), performance improved as a function of trials at 9 months (ANOVA F(2, 91) = 9.7, p < 0.001, Huynh–Feldt correction, Fig. 3C) and 12 months of age (ANOVA F(2, 68) = 3.72, p < 0.05, Huynh–Feldt correction, Fig. 3C). Again, there was no significant difference between transgenic and wild-type animals at 9 months (ANOVA F(2, 91) = 1.10, p = 0.34, Huynh–Feldt correction) and 12 months of age (ANOVA F(2, 68) = 1.15, p = 0.32, Huynh–Feldt correction; Fig. 3C).

Finally, a probe trial was conducted at the end of testing to assess the amount of time spent in the target quadrant. At 9 months (t(39) = 0.43, p = 0.67; Fig. 3D) and 12 months of age (t(38) = 0.30, p = 0.77; Fig. 3D) there was no significant difference in time spent in target quadrant between transgenic LRRK2R1441G rats and their wild-type littermates (Fig. 3D).

In summary, there were no cognitive deficits detectable in LRRK2R1441G rats with the Morris water maze.

Paraquat vulnerability in aged LRRK2R1441G transgenic rats

In order to assess gene environment interactions in PD, vulnerability to Paraquat was tested in aged LRRK2R1441G transgenic rats at approximately 14–16 months of age. Transgenic and wild-type rats were exposed to acute Paraquat at a subtoxic dose and rearing behavior was recorded. Animals received two injections of Paraquat (n = 8 wild-type and 11 LRRK2R1441G) or saline (n = 8 wild-type and 10 LRRK2R1441G) and rearing behavior was measured immediately after each injection, 24 h after each injection and ten days post injection (Fig. 4). A significant reduction in rearing behavior over testing time points was noted in all animals (ANOVA F(3, 83) = 9.30, p < 0.001, Greenhouse-Geisser correction). As expected, there was no overall main effect of drug, as a low dose of Paraquat was used in this experiment (ANOVA F(3, 83) = 2.50, p = 0.83, Greenhouse-Geisser correction). There was no interaction between time, drug and genotype (ANOVA F(3, 83) = 0.30, p = 0.83, Greenhouse-Geisser correction). Therefore, LRRK2R1441G transgenic rats did not show increased vulnerability to Paraquat at the dose employed. In addition, univariate ANOVA analysis was conducted at each testing point. Immediately after injection 1, Paraquat exposed groups displayed significantly reduced rearing movements (ANOVA F(1, 32) = 5.51, p < 0.05), however the transgenic animals did not significantly differ from wild-type controls (ANOVA F(1, 32) = 0.033, p = 0.857). This effect was lost 24 h post injection 1, and no significant difference in rearing behavior was noted between groups exposed to Paraquat or saline (ANOVA F(1, 28) = 1.81, p = 0.189). Immediately following injection 2, again, a significant main effect of drug was again noted (ANOVA F(1, 28) = 8.45, p < 0.05), but no significant difference between transgenic and wild-type controls (ANOVA F(1, 28) = 0.23, p = 0.64). At 24 h later, there was no difference between groups exposed to Paraquat or saline (ANOVA F(1, 28) = 2.73, p = 0.06). Ten days post injection two, there was also no significant main effect of drug (ANOVA F(1, 28) = 1.10, p = 0.30) or genotype and drug interaction (ANOVA F(1, 28) = 0.37, p = 0.55).

Figure 4 Vulnerability to paraquat.

Rearing behavior following acute Paraquat or saline exposure was measured in LRRK2R1441G and wild-type rats. All animals were tested immediately after a first injection (injection 1), 24 h after injection 1, immediately after injection 2, 24 h after injection 2, and ten days after the last injection. There was a reduction in rearing movements in the Paraquat-exposed groups immediately after injections 1 and 2, but no significant difference in drug reactions between genotypes. Ten days following injections, there was no more drug effect on rearing behavior in any genotype. N = 8 wt × PQ, n = 10 LrrK2 × PQ, n = 8 wt × saline, n = 10 wt × PQ (one LrrK2 animal that died within this testing period was eliminated from this data).

In summary, Paraquat had a short term effect on rearing behavior immediately following injections, but this effect was not selective for LRRK2R1441G rats, and was not detectable ten days post injection.

Animal survival

As animals aged beyond the 12 month time point, and specifically during the Paraquat testing, 2 wt animals and 10 transgenic LRRK2R1441G rat spontaneously died. Kaplan–Meier curves and a log rank (Mantel-Cox) test were conducted to investigate survival estimates in transgenic LRRK2R1441G and wild-type rats until all surviving animals were sacrificed to harvest tissue for qRT-PCR (Fig. 5). While there was no statistical significant difference in survival time between transgenic and wild-type animals up to that point (X2(1) = 2.65, p = 0.10), it might have become apparent if animals would have been allowed to survive for a longer time period. We chose to sacrifice the remaining animals at 18 months of age in order to ensure that we can harvest tissue for quantitative RT-PCR analysis of gene expression levels. Interestingly, post mortem analysis of rats that did not survive revealed different hemopoietic tumors in all transgenic animals, but only one wild-type animal as probable cause of death.

Figure 5 Survival curve.

Survival rates were plotted for LRRK2R1441G (M = 17.91, SEM = 0.38) and WT rats (M = 18.75, SEM = 0.17). A log rank (Mantel-Cox) test revealed no significant differences in survival between transgenic and wild-type animals.

Histology and qRT-PCR

In order to assess potential neuro-degeneration in the substantia nigra, we performed immunostaining of dopaminergic neurons using a tyrosine hydroxylase antibody (Fig. 6A), and of cell nuclei using Hoechst 33342 staining at age stages of 3, 6, and 18 months (Fig. 6B). Stereological quantification of TH staining revealed no significant age and genotype interaction in the amount of staining in the substantia nigra (ANOVA F(2, 2) = 0.022, p = 0.979; Fig. 6C). Analysis of the Hoechst 33342 stain revealed no differences in the number of stained nuclei in the substantia nigra, nor an apparent condensation or fragmentation of nuclei at the older age stage that would indicate a higher level of apoptosis (ANOVA F(1, 2) = 0.00, p = 1.00; Fig. 6D).

Figure 6 Substantia nigra histology.

(A) Tyrosine hydroxylase immunostaining of the substantia nigra in brain slices of 3, 6, and 18 months old rats. Slices were counterstained with Nissl staining. (B) Hoechst 33342 stained sections of 3, 6, and 18 month old rats show cell nuclei in the substantia nigra. There are no pyknotic or fragmented cell nuclei that would indicate a higher rate of apoptotic cell death in old or transgenic animals. (C) Quantification of DAB stained area of three sections per animal. Tyrosine hydroxylase immunostained area was not significantly different between genotypes. (D) The number of cell nuclei within a defined window does not differ between age stages, indicating that the total number of cells does not considerably change despite the smaller area of DAB staining in older animals. Error bars refer to standard deviation between three brain section analysed per brain by two different scorers each.

Additionally, we quantified staining for tyrosine hydroxylase (TH) in the dorsal striatum, the main termination area of dopaminergic projections from the substantia nigra (Fig. 7A). We found no difference in DAT expression between genotypes (p = 0.864), age groups (p = 0.258), or interactions of genotype and age group (ANOVA F(2, 2) = 0.022, p = 0.997; Fig. 7B). We also quantified the expression of the dopamine transporter (DAT) in the dorsal striatum. ANOVA revealed a significant effect of age group on DAT expression (ANOVA F(2, 2) = 21, p = 0.048; Figs. 7C and 7D), but no effect of genotype or interaction of genotype with age group.

Figure 7 Striatum histology.

(A) Dopamine transporter (DAT) immunoreactivity in the striatum of 3 and 9 month old wt and LRRK2R1441G rats. The squares indicate the regions that were quantitatively analyzed. (B) Quantitative analysis of DAT staining in the striatum revealed a significant reduction of DAT expression in older animals, but no difference between genotypes. (C) Expression of tyrosine hydroxylase (TH) in the striatum of the same rats. (D) Quantitative analysis of TH expression revealed no difference between age groups and genotypes. Error bars refer to standard deviation between three brain section analysed per brain by two different scorers each.

Initial qRT-PCR results have indicated an expression of 8–10 copies of the transgene in these rats, and genotyping of rats tested here had confirmed a genomic presence of human LRRK2 in all transgenic animals (see Fig. S1). We sacrificed the 13 surviving transgenic rats at the end of our study in order to repeat qRT-PCR analysis of gene expression. The results revealed a low expression of human LRRK2 in the substantia nigra, cortex, hippocampus, liver or kidney, that was below detection level (see Table 1), indicating that the gene expression levels had been down regulated in the surviving animals.

Table 1 qRT-PCR.

Relative expression of the human LRRK2R1441G RNA in the substantia nigra of transgenic animals. A wild-type sample served as negative control and a sample of a LRRK2G2019S rat as positive control. Expression was normalized to the expression of the housekeeping gene GAPDH.

Animal ID	Genotype	DOB	Tissue	LRRK2 expression	GAPDH expression	
				Mean	SD	Mean	SD	
101		7/26/2012	Hippocampus	–	–	17.808	0.203	
			Substantia Nigra	–	–	18.217	0.287	
			Cortex	–	–	21.133	0.212	
			Kidney	–	–	19.127	0.142	
			Liver	–	–	18.912	0.096	
102		7/26/2012	Hippocampus	–	–	18.730	0.169	
			Substantia Nigra	–	–	19.639	0.025	
			Cortex	–	–	17.477	0.581	
			Kidney	–	–	18.105	0.372	
			Liver	–	–	18.790	0.161	
104		7/26/2012	Hippocampus	–	–	18.897	0.928	
			Substantia Nigra	–	–	21.802	4.886	
			Cortex	–	–	19.578	0.121	
			Kidney	–	–	19.308	0.360	
			Liver	–	–	22.004	0.194	
102		7/26/2012	Hippocampus	–	–	18.730	0.169	
			Substantia Nigra	–	–	19.639	0.025	
			Cortex	–	–	17.477	0.581	
			Kidney	–	–	18.105	0.372	
			Liver	–	–	18.790	0.161	
104		7/26/2012	Hippocampus	–	–	18.897	0.928	
			Substantia Nigra	–	–	21.802	4.886	
			Cortex	–	–	19.578	0.121	
			Kidney	–	–	19.308	0.360	
			Liver	–	–	22.004	0.194	
303		8/5/2012	Hippocampus	–	–	20.458	0.347	
			Substantia Nigra	–	–	20.347	0.961	
			Cortex	–	–	19.271	0.261	
			Kidney	–	–	18.998	1.423	
			Liver	–	–	21.391	0.311	
305		8/5/2012	Hippocampus	–	–	19.391	0.287	
			Substantia Nigra	–	–	20.340	1.455	
			Cortex	–	–	19.271	0.261	
			Kidney	–	–	18.901	0.102	
			Liver	–	–	21.164	0.351	
404		9/16/2012	Hippocampus	–	–	19.710	0.153	
			Substantia Nigra	–	–	20.246	0.051	
			Cortex	–	–	19.144	0.567	
			Kidney	–	–	20.316	0.669	
			Liver	–	–	23.021	3.470	
406		9/16/2012	Hippocampus	–	–	21.827	5.225	
			Substantia Nigra	–	–	20.121	0.083	
			Cortex	–	–	18.711	0.251	
			Kidney	–	–	19.768	0.282	
			Liver	–	–	22.133	0.081	
502		9/25/2012	Hippocampus	–	–	18.933	0.103	
			Substantia Nigra	–	–	20.223	0.550	
			Cortex	–	–	20.541	0.069	
			Kidney	–	–	19.134	0.029	
			Liver	–	–	21.181	0.335	
503		9/25/2012	Hippocampus	–	–	23.215	0.079	
			Substantia Nigra	–	–	24.910	0.151	
			Cortex	–	–	23.557	0.258	
			Kidney	–	–	25.013	3.874	
			Liver	–	–	20.843	0.110	
506		9/25/2012	Hippocampus	–	–	18.746	0.283	
			Substantia Nigra	–	–	18.474	0.085	
			Cortex	–	–	21.564	0.106	
			Kidney	–	–	28.604	0.233	
			Liver	–	–	21.927	0.899	
703		9/29/2012	Hippocampus	–	–	21.522	0.130	
			Substantia Nigra	–	–	34.326	0.436	
			Cortex	–	–	21.379	0.127	
			Kidney	–	–	26.567	0.194	
			Liver	–	–	23.392	0.092	
Pos. C		n/a	Human tissue	15.452	0.324	21.522	0.130	

Discussion

Animal models are critical tools for not only understanding LRRK2 pathology, but PD aetiology and pathogenesis. In particular, models which carry PD related mutations on the human LRRK2 gene can provide insight into LRRK2 function and the mechanisms by which it mediates PD related dysfunction. Recently, bacterial artificial chromosome (BAC) LRRK2 transgenic rodent models have been generated which potentially allow for investigating the link between genetic mutations and development of PD. Here we have characterized newly available transgenic LRRK2R1441G BAC rats. Unfortunately, these rats do not display PD related motor or cognitive deficits by 12 months of age, nor do they show increased vulnerability to Paraquat exposure as compared to age-matched wild-type controls.

An important limitation of the use of these animals is that there were no more detectable levels of transgene expression in the aged LRRK2R1441G transgenic rats that survived until the end of study. While all transgenic rats did display genomic human LRRK2R1441G in genotyping PCR results, this gene seems to have become silenced at least in the surviving animals despite the fact that a BAC approach was used. BAC constructs are large genomic inserts and due to their size (∼200 kb) allow the inclusion of endogenous regulatory elements. However, the loss of transgene expression noted in this model suggests that transgene insertion, even with the BAC approach, was not stable. It is a general limitation of the transgenic approach that—due to random integration of the transgene—positional effects can occur. Position effects refer to the phenomenon whereby transgene expression is influenced by the site of integration. Position effects can include lack of transgene expression and extinction of transgene expression in successive generations.

Accordingly, post-mortem analysis conducted on rats which did not survive until the end of our study showed pathological morphologies consistent with hemopoietic tumors in LRRK2R1441G. A recent study by Ruiz-Martinez et al. (2014) reported a higher prevalence of hematological cancers in patients carrying the LRRK2R1441G mutation. The slightly higher mortality and the occurrence of hemopoietic tumors in all transgenic rats that died before the end of our study indicate that gene silencing might not have happened in all animals to the same extend.

Although to our knowledge no previous studies have examined PD phenotypes in LRRK2R1441G transgenic BAC rats, our results are consistent with a recent study where transgenic LRRK2 BAC mice failed to consistently reproduce cardinal features of PD. Although Li et al. (2009) originally reported age-dependent, L-DOPA responsive, progressive motor dysfunction in LRRK2R1441G BAC mice by 10 months of age, subsequent studies in transgenic mice have been unable to reproduce these results. Bichler et al. (2013) did not observe gross motor dysfunction in transgenic LRRK2R1441G BAC mice by 10 months of age, although modest motor deficits were observed in advanced age (21 months of age). These mice also did not display cognitive symptoms of PD, including depression and anxiety-like behaviour and impaired learning and memory (Bichler et al., 2013). However, gastrointestinal dysfunction, a common early non-motor symptom of PD was noted in these mice by 6 months of age (Bichler et al., 2013). Dranka et al. (2013) did not report gross motor abnormalities in LRRK2R1441G BAC mice, although by 16 months of age, these mice did display some deficits in motor coordination, an early symptom of PD. Similarly, other classic transgenic LRRK2 mouse models over-expressing mutant LRRK2 have mostly failed to show PD specific gross motor deficits (Li et al., 2009; Li et al., 2010; Melrose et al., 2010) despite mild nigral neurodegeneration (Ramonet et al., 2011). An interesting approach was pursued by Tong and colleagues (2009) who created a LRRK2 R1441C knock-in mouse, which is a model most closely mimicking the human PD situation without LRRK2 overexpression. These mice did also not show any PD symptoms or nigrostriatal degeneration, but they had altered DA transmission that became evident upon injection of amphetamine or the D2 receptor agonist quinpirole. This impairment in stimulated dopamine transmission was suggested to be a pathogenic precursor preseding neurodegeneration in PD brains.

Rodents as PD models

Parkinson disease is characterized by slow and progressive degeneration of the substantia nigra and disruption of basal ganglia circuitry with advancing age. While neurotoxin models which display disease outcomes have had some success, the progressive nature of the disease calls for a genetic model with a progressive phenotype. The relatively short life span of rodents as compared to humans, however, may make them imperfect models for studying genetic forms of PD. A healthy laboratory rat can survive for maximally 2–3 years while LRRK2 mediated PD is not apparent in humans until after 65 years of age. Indeed, despite the discovery of many genetic mutations, the greatest known risk factor for PD continues to be advanced age. The rats used in this study were up to 12–16 months of age, which is considered ‘advanced’ age for rats (Quinn, 2005), but this may not have been sufficient to reproduce PD related phenotypes, even if some of them may have expressed the transgene to a considerable extend. Some researchers have argued that extremely high levels of transgene expression can induce neurological phenotypes within the life span of a rodent. Ramonet et al. (2011) tested four different transgenic mouse lines, expressing either the G2019S mutation or the R1441C mutation and observed neuronal loss in advanced age in only one line, in which transgene expression was more than 300% greater than the level of endogenous LRRK2. However, in this case non-physiological levels of transgene expression are used which, while producing PD phenotypes, may limit our ability to translate animal research to human disease.

In addition, the transgene is constitutively expressed in most rodent models, including our rats, so they were able to develop compensatory mechanisms which potentially counteracted the toxic effects of mutated LRRK2. While LRRK2 mutations in familiar PD are constitutively expressed as well, the over-expression of mutated LRRK2 might trigger the compensatory effects. Zhou et al. (2011) found no behavioural phenotypes in rats that constitutively expressed mutated LRRK2G2019S despite high gene expression, however, they did observe abnormal motor activity in rats with conditional adult onset expression of mutated LRRK2G2019S. Their results suggest that rats are able to accommodate LRRK2’s toxic effects when it is constitutively expressed. With this in mind, it might be more favourable to consider a conditional transgenic animal model and/or a viral mediated gene transfer approach instead of the BAC approach. With the viral mediated gene transfer approach, the mutated LRRK2 gene can be delivered to rodents through a viral vector in adulthood, thus bypassing the development of compensatory effects. This approach also allows researchers to target specific neuronal populations, such as susbtantia nigra dopaminergic neurons, to better model the degeneration seen in PD. Another advantage of the viral mediated gene transfer approach is that it allows researchers to correlate transgene dosage (by modulating copy number of transgene) with phenotype severity. Studies using this approach have already had success in recapitulating cardinal features of PD. Lee et al. (2010) created a mouse model of PD expressing human LRRK2G2019S using herpes simplex virus (HSV), which displayed 50% degeneration of nigral dopaminergic neurons. In addition a rat model generated by Dusonchet et al. (2011) expressing human LRRK2G2019S showed a 20% reduction in dopaminergic neurons. Tsika et al. (2014) developed conditional transgenic mice that selectively express human LRRK2R1441C in dopaminergic neurons from the endogenous murine ROSA26 promoter. The animals showed no neurodegeneration or locomotor symptoms, but altered dendritic and nuclear morphology in dopaminergic neurons upon aging, indicating the onset of pathogenic processes. While the viral overexpression of mutated LRRK2 also has its clear limitations due to the non-physiological doses of LRRK2, viral models may allow us to recapitulate some of the neurodegeneration processes observed in PD patients, which has so far been difficult to show in other models.

In summary, we here characterized transgenic BAC rats carrying the human LRRK2R1441G mutation in terms of motor function, sensorimotor gating, cognitive function and dopaminergic cell loss in the substantia nigra. Transgenic LRRK2R1441G rats did not stably express the transgene to a satisfactory extend, they didn’t display any symptoms reminiscent of Parkinson disease by 12 months of age, and they did not show increased vulnerability to the environmental toxin Paraquat. Overall, our results indicate that these transgenic BAC LRRK2R1441G rats are not a viable model of Parkinson disease.

Supplemental Information

Supplemental Information 1 Raw data

Click here for additional data file.

Supplemental Information 2 Genotyping example for wild-type and transgenic animals

Genotype results show strong positive bands at 387 bp, indicating genomic presence of the human LRRK2 transgene.

Click here for additional data file.

Additional Information and Declarations

Competing Interests

Author Contributions

Animal Ethics

The authors declare there are no competing interests.

Komal T. Shaikh conceived and designed the experiments, performed the experiments, analyzed the data, wrote the paper, prepared figures and/or tables.

Alvin Yang and Ekaterina Youshin performed the experiments.

Susanne Schmid conceived and designed the experiments, wrote the paper, prepared figures and/or tables, reviewed drafts of the paper, handling the manuscript.

The following information was supplied relating to ethical approvals (i.e., approving body and any reference numbers):

University of Western Ontario Animal Use Subcommittee, protocol 2011-077.

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
