# Peer review of "Transgenic LRRK2R1441G rats–a model for Parkinson disease?"

_PeerJ, doi:10.7717/peerj.945_

## Round 0.1 · original submission · Major Revisions

Please consider the reviews and revise the manuscript accordingly.

Reviewer 1 ·

Basic reporting

In the introduction the authors have stated that the R1441G mutation increases kinase activity and provided a reference to a clinical study on the penetrance of the G2019S mutation by Healy et al. It is debatable whether the R1441G mutation increases kinase activity with a number of papers concluding yes or no depending on how activity is measured. Better references and discussion surrounding this should be included. It would be better still to directly assess LRRK2 R1441G activity in the model used.

Experimental design

No comments

Validity of the findings

The authors found that their transgenic mice fell into 2 groups. Mice that survived to end and showed no phenoype, and mice that seem to die earlier from tumours. The mice that survived with no phenotype or tumours reportedly had no expression of the trangsene, leading the authors to conclude that the transgene had been silenced. More evidence needs to be provided to show that these are indeed transgenic rats and not wild-type rats mislabelled. Could the rats be mixed up during behavioural testing? Were the rats genotyped again at the end of the study? Is there a second genotyping method to confirm the results (southern blot?)? Western blot analysis should be performed to see the levels of overexpressed protein rather than inferring from PCR. Western blot data comparing the expression levels of LRRK2 at the start and end of the study should be provided to conclude that the transgene has been silenced. What happens to the analysis if this group of rats is removed?

The discussion debates the pro and cons of transgenic overexpressing models and suggests that viral gene transfer approaches may be better for overexpressing LRRK2. It should be noted that overexpression models are not physiological as LRRK2 is not overexpressed in human Parkinson’s disease. The discussion should compare with endogenous knockin models (ie. R1441C mice) as they more closely mimic the human disease. A recent paper showing how variable viral transfer models are could be included (Neurobiol Dis. 2014 Nov;71:345-58.).

Additional comments

The manuscript by Shaikh et al describes the characterisation of a commercially available transgenic rat model of the Parkinson’s disease causing LRRK2 protein. In contrast to a similar mouse model generated by another lab they found no defects in motor function, cognition or neurodegeneration. It is a negative study but the behavioural testing and analysis seem to have been performed well.

Reviewer 2 ·

Basic reporting

No comments

Experimental design

No comments

Validity of the findings

No comments

Additional comments

In this paper Shaikh and collaborators characterized the overexpression of the LRKK2-R1441G gene in the rat as an animal model for PD. They tested their deficits in motor function, sensorimotor gating, and higher cognitive function at different periods of time (3, 6, 9 and 12 months). They also injected the rats with Paraquat to test a different vulnerability to this toxin. These rats do not display any PD related motor or cognitive deficits and they do not show any different vulnerability to Paraquat.

The novelty of this paper is not high. Overall, LRRK2 rodent models display mild or not functional disruption of the nigrostriatal dopaminergic system, including overexpression of G2019S-LRRK2, so this is not surprising. While the experimental data presented here add more evidence to the fact that the transgenic LRRK2 animal models are not good models for studying the pathology of PD, there are several technical issues that limit the interpretation of the data presented.

One of the main limitations of this study is that there were no detectable levels of transgene expression in the rats at the end of study. This can suggest that the transgene insertion was not stable at all, as the authors pointed out. In addition, the transgene was constitutively expressed here, so the rats could develop mechanisms to counteract the possible effects of LRRK2. One of the main troubling observations in genetic models is the inconsistent phenotype among the lines with the same mutations.

Regarding the use of Paraquat, rats are resistant to MPTP, but the effect of Paraquat in the nigrostriatal dopaminergic system is somewhat ambiguous. This model shows contradictory results, variable cell death and loss of DA when is used in rats. Moreover, the authors used a sub-toxic dose, so there is the possibility that this dose is not sufficient to cause any lesion even in more susceptible rats. Can the authors provide any information using a toxic dose?

The battery of motor and cognitive test used to characterize these animals seems correct and well defined, but the method used to analyzed the substantia nigra and the dopaminergic lesion is either not correct or is not well explained. In page 12 the authors say that “Cells were stained for TH and then quantified using ImageJ” while in page the authors refer to “Stereological quantification of staining” and in Figure 6 they show “Quantification of DAB stained area of three sections per animal” What does “Stereological quantification of staining” mean? This should be clarified. The gold standard for this type of assessment is stereology. I would like to see also the terminal density at the striatum or the dopamine levels of these rats. This is really important because the loss or the lack of loss of dopamine in the striatum or cells in the substantia nigra is crucial.

Why did the authors counterstain with Nissl staining if they don’t show any data?

The authors should include the number of animals used for each experiment in the figure legends. For example, they said that “One wild-type and one LRRK2R1441G rat of 3 months and of 6 months of age and two rats per genotype of 18 months of age were used for hystological analysis” (Page 11) but they showed error bars in Figure 6, how is it possible? It is confusing.

---

## Round 0.2 · accepted · Accept

Congratulations on your work.